# Molecular Dynamics and In Vitro Studies Elucidating the Tunable Features of Reconfigurable Nanodiscs for Guiding the Optimal Design of Curcumin Formulation

**DOI:** 10.3390/bioengineering11030245

**Published:** 2024-02-29

**Authors:** Yongxiao Li, Wanting Xu, Xinpei Wang, Ruizhi Lai, Xiaohui Qiu, Zekai Zeng, Zhe Wang, Junqing Wang

**Affiliations:** 1School of Pharmaceutical Sciences, Shenzhen Campus of Sun Yat-sen University, Shenzhen 518107, China; liyx356@mail2.sysu.edu.cn (Y.L.); xuwt27@mail2.sysu.edu.cn (W.X.); wangxp39@mail2.sysu.edu.cn (X.W.); qiuxh27@mail2.sysu.edu.cn (X.Q.); 2Department of Pathology, The Eighth Affiliated Hospital, Sun Yat-sen University, Shenzhen 518033, China; lairzh3@mail2.sysu.edu.cn (R.L.); zengzk3@mail2.sysu.edu.cn (Z.Z.)

**Keywords:** NDs, curcumin, apoA-I, lipid bilayer, membrane, anticancer

## Abstract

In this study, we advance our exploration of Apolipoprotein A-I (apoA-I) peptide analogs (APAs) for their application in nanodisc (ND) assembly, focusing on the dynamic conformational characteristics and the potential for drug delivery. We explore APA-ND interactions with an emphasis on curcumin encapsulation, utilizing molecular dynamic simulations and in vitro assessments to evaluate the efficacy of various APA-ND formulations as drug carriers. The methodological approach involved the generation of three unique apoA-I α-11/3 helical mimics, resulting in fifteen distinct APAs. Their structural integrity was rigorously assessed using ColabFold-AF2, with particular attention to pLDDT and pTM scores. Extensive molecular dynamics simulations, covering 1.7 μs across 17 ND systems, were conducted to investigate the influence of APA sequence variations on ND stability and interactions. This study reveals that the composition of APAs, notably the presence of Proline, Serine, and Tryptophan, significantly impacts ND stability and morphology. Oligomeric APAs, in particular, demonstrated superior stability and distinct interaction patterns compared to their monomeric counterparts. Additionally, hydrodynamic diameter measurements over eight weeks indicated sequence-dependent stability, highlighting the potential of specific APA configurations for sustained colloidal stability. In vitro study successfully encapsulated curcumin in [AA]_3_/DMPC ND formulations, revealing concentration-dependent stability and interaction dynamics. The findings underscore the remarkable capability of APA-NDs to maintain structural integrity and efficient drug encapsulation, positioning them as a promising platform for drug delivery. The study concludes by emphasizing the tunability and versatility of APA-NDs in drug formulation, potentially revolutionizing nanomedicine by enabling customized APA sequences and ND properties for targeted drug delivery.

## 1. Introduction

Curcumin, a polyphenolic compound derived from the turmeric plant (*Curcuma longa*), has gained significant scientific attention for its potential therapeutic properties [1,2]. Historically, curcumin has been used in Ayurvedic and traditional Chinese medicine for centuries, primarily for its anti-inflammatory and antioxidant properties [3]. In recent years, the bio-applications of curcumin have expanded significantly, with research exploring its potential in treating various diseases [4]. These include neurodegenerative disorders like Alzheimer’s and Parkinson’s disease [5,6], cardiovascular diseases [7], diabetes [8], arthritis [9], and several types of cancer [10,11]. Its mechanism of action is complex and multifaceted, involving the modulation of a diverse array of molecular targets, which is crucial in various pathological processes [12]. Curcumin has been shown to exert profound anti-inflammatory effects by downregulating inflammatory cytokines such as TNF-α and interleukins [13] and by inhibiting key enzymes like cyclooxygenase-2 (COX-2) and inducible nitric oxide synthase (iNOS) [14,15], pivotal in the inflammatory cascade. Additionally, curcumin significantly impacts transcription factors, notably inhibiting nuclear factor kappa B (NF-κB) [16], thus regulating genes involved in inflammation, cell proliferation, and survival. Its interaction with growth factors, particularly the inhibition of vascular endothelial growth factor (VEGF) [17], underscores its potential to impede angiogenesis (essential for tumor growth). Moreover, curcumin induces cell cycle arrest and apoptosis in cancer cells, mediated through the modulation of cell cycle proteins and apoptosis-related pathways [18]. Recent insights also reveal its role in epigenetic modifications, which may have far-reaching implications in cancer therapy and diseases where epigenetic factors are influential [19]. The extensive scope of curcumin biological interactions emphasizes its potential as a therapeutic agent, necessitating further research to fully harness its medicinal properties. Despite its promising therapeutic benefits, the clinical application of curcumin is hindered by its poor bioavailability, characterized by low water solubility and rapid metabolic degradation [20]. This results in insufficient systemic absorption and limited tissue distribution, presenting a significant challenge for its therapeutic application. To address these pharmacokinetic limitations, recent advances in nanotechnology have been instrumental in the development of numerous nano-formulations. Liposomal encapsulation [21], polymeric nanoparticles [22], and micellar systems [23] have emerged as promising strategies to enhance the solubility, stability, and targeted delivery of curcumin.

Recently, lipid bilayer nanodiscs (NDs) have emerged as a novel and promising alternative in the field of drug delivery systems [24], particularly for hydrophobic compounds like curcumin. Pioneering this approach, Ghosh et al. (2011) leveraged NDs to enhance curcumin’s solubility and target its release, marking a significant advancement in its therapeutic application [25]. Subsequent research by Singh et al. (2011) corroborated the efficacy of curcumin-loaded NDs, especially in the treatment of mantle cell lymphoma (MCL) and other cancer forms [25]. Further insights by Ghosh and Ryan (2014) revealed that the coupling of curcumin NDs with ApoE notably augments curcumin absorption in glioblastoma multiforme cells, thereby amplifying its biological efficacy [26]. Recently, Evans et al. reported on the development of curcumin-loaded nanolipoprotein particles (cNLPs) as an ND approach to improve curcumin bioavailability for administration as a radioprotective and/or radiomitigative agent against ionizing radiation exposures in non-cycling cells or as a radiosensitizing agent for actively dividing cell populations such as tumors [27]. This series of studies underscores the potential of NDs as a transformative vehicle for curcumin delivery, highlighting the versatility and potential of NDs in medical research and treatment strategies.

Compared to conventional delivery systems such as liposomes, polymeric nanoparticles, and micelles, NDs offer distinct features. Primarily, they mimic natural cell membranes and provide a stable and biocompatible environment for encapsulating lipophilic compounds like curcumin, ensuring a controlled and sustained release [28]. Furthermore, they exhibit unique, small discoid structures (10–30 nm) that can easily penetrate different tissues without being quickly removed by circulation [29]. It is believed that NDs were able to better penetrate tumors due to their more elastic and flexible architecture and performed significantly better than liposomes for long-term tumor remission [30] and T-cell priming for antitumor immunity [31]. In terms of biocompatibility and toxicity, NDs offer a safer alternative. NDs are generally composed of biocompatible lipid materials, significantly reducing the risk of adverse reactions and making them more suitable for long-term therapeutic use. Their ability to be engineered for specific tissue targeting further amplifies their therapeutic potential, allowing for more precise and effective treatment strategies [24,32].

We previously reported on the development of reconfigurable peptide analogs of Apolipoprotein A-I (apoA-I) to study nanodisc (ND) assembly [33]. Our findings revealed that apoA-I peptide analogs (APAs) significantly influence ND assembly in terms of APA oligomeric states, sequence composition, and lipid-to-APA ratio. Molecular simulations and morphological validations showed that longer APAs with more tandem repeats yield higher particle size homogeneity. Proline was identified as dispensable for APA-ND formation, but its presence influenced the size of NDs. Overall, our study provides insights into the rational design of apolipoproteins for ND engineering. In another study, we developed an osteoblast-targeting peptide fused with APA, which can assemble into NDs and is loaded with a lysine-specific demethylase 6B (KDM6B) inhibitor for bone and macrophage dual-targeted delivery, and explored the role of KDM6B in osteogenic differentiation of mesenchymal stem cells (MSCs) and its interaction with macrophages [33]. Recently, we explored the ND formulation of Doxorubicin (DOX), a potent cancer drug, using molecular dynamics (MD) simulations [34]. Our study revealed differences in drug release profiles and conformational stability between free DOX and lipid-conjugated DOX-prodrug ND formulations. These findings underscore the versatility of APA-NDs and their potential use in bio-application.

In the present study, we delve deeper into the adjustable dynamic conformational characteristics of APA-NDs (Apolipoprotein A-I Mimetic Peptide NDs) and explore the formulation of curcumin within this APA-ND framework using MD simulations. This approach is driven by the need to understand intricate interactions at a molecular level. Additionally, we conduct comparative studies on the in vitro biological activity across various APA-ND formulations, which are aimed at unraveling the nuances of APA-NDs as a delivery mechanism for curcumin, thereby contributing to the optimization of their therapeutic potential.

## 2. Materials and Methods

### 2.1. Reagents

The following reagents were acquired from various suppliers: 1,2-dimyristoyl-sn-glycero-3-phosphocholine (DMPC) from Avanti Polar Lipids (Birmingham, UK), Sodium Cholate from TCI AMERICA (Portland, OR, USA), Bio-Beads SM-2 from Bio-Rad (Hercules, CA, USA), DiOC14(3) from Biotium (Shanghai, China), CellLight™ Early Endosomes-RFP (BacMam 2.0) from ThermoFisher (Shanghai, China), Annexin V-FITC/PI apoptosis kit from ES Science (Hangzhou, China), and cell culture reagents from Gibco (Grand Island, NY, USA). Other necessary reagents were sourced from Sigma (Shanghai, China).

### 2.2. APA Peptide Synthesis

All APA peptides were synthesized using a solid-phase method [35], employing 9-Fluorenylmethyloxycarbonyl (Fmoc) as the protective group. The peptides were detached from the resin with a mixture of trifluoroacetic acid, thioanisole, 1,2-ethyldithiol, phenol, and water in the proportions of 87.5/5/2.5/2.5/2.5. Subsequent purification was performed using the Waters Alliance 2695 HPLC System with a 2487 Dual Absorbance Detector (Water Alliance, Federal Way, WA, USA), utilizing a ZORBAX SB-C18 (Agilent, Santa Clara, CA, USA) column and a gradient of water and acetonitrile (H_2_O-ACN) with 0.1% trifluoroacetic acid (TFA). The peptides’ retention times were measured via HPLC. Identification was carried out using Water ZQ electrospray ionization mass spectrometry (Water Alliance, Federal Way, WA, USA). All peptides were prepared in lyophilized powder form and stored at −20 °C until needed.

### 2.3. Empty ND, Dye-Loaded ND, and Curcumin-Loaded ND (Cur-ND) Preparation

The process for assembling these NDs began by dissolving DMPC in chloroform. This mixture was then dried using a nitrogen flow and subsequently placed in a vacuum desiccator for over 4 h. The dried phospholipid was hydrated using MSP buffer (containing 20 mM Tris-HCl, pH 7.5, 100 mM NaCl, and 0.5 mM EDTA), to which sodium cholate (NaCh) was added, followed by the addition of MSP towards the end. The final concentrations of lipid, NaCh, and MSP varied under different conditions, as detailed in the main text. The assembly mixture was then incubated for 2 h at room temperature. Following this, Bio-Beads SM-2 (Bio-Rad) were added at a concentration ranging from 0.5 to 1 g per milliliter of the assembly mixture, and the mixture was shaken for 3 h at room temperature. The Bio-Beads were then removed by filtration through a 0.22-μm filter. The ND preparations underwent further purification using size exclusion chromatography (SEC) on a Superdex200 Increase 10/300 GL column (GE Healthcare, South Plainfield, NJ, USA) at a flow rate of 0.5 mL/min. Both fluorescent dye (DiO)-loaded ND and Cur-ND were prepared following the same procedure. The concentrations of DiO and curcumin used were 25 μM and 0.3 mM, respectively.

### 2.4. Dynamic Light Scattering (DLS)

The measurement of ND particle sizes was conducted using DLS with a Nano-ZS90 Zetasizer (Malvern Instruments, Worcestershire, UK). This analysis encompassed four distinct batches for each formulation, conducted at a controlled temperature of 25 °C, with each batch undergoing nine separate measurement iterations.

### 2.5. Entrapment Efficiency (EE%) and Loading Capacity (LC%) of Cur-NDs

EE% and LC% of Cur-NDs, a curcumin standard curve was first established. This curve was created by dissolving free curcumin in methanol at concentrations of 8, 10, 20, 40, 60, 80, 100, and 200 μM and measuring the absorbance at an optical density (OD) of 428 nm. For the Cur-ND samples, any unencapsulated curcumin was removed, and methanol was then added to ensure complete dissolution of the curcumin encapsulated within the NDs. The EE% and LC% values for the Cur-ND were then calculated by comparing their absorbance values to the established curcumin standard curve.

### 2.6. Cells Lines and Cell Culture

The cell lines used in this study included U-87 MG (human glioblastoma), HepG2 (hepatoma), and NCM460 (normal colonic epithelial). These cells were cultured in Dulbecco’s Modified Eagle Medium (DMEM) enriched with 10% fetal bovine serum (FBS), along with 100 µg/mL streptomycin and 100 U/mL penicillin. They were maintained as monolayers and incubated at 37 °C under an atmosphere containing 5% CO_2_.

### 2.7. MTT Assay

Human cell lines U-87 MG, HepG2, and NCM460 were seeded in quadruplicate in 96-well plates, with 5000 cells per well, 24 h before drug treatment. To evaluate the cytotoxicity of empty NDs, these cells were exposed to ND concentrations of 0, 0.05, 0.10, 0.20, 0.40, 0.80, 1.00 mg/mL for an additional 24 h. After this period, MTT solution (5 mg/mL) was added to each culture and incubated for 4 h. The media were then removed from the wells, and DMSO was added to dissolve the MTT formazan product for analysis. Cell viability was assessed by measuring the absorbance at an OD of 490 nm using a SpectraMax i3x Multi-Mode Detection Platform (Molecular Devices, San Jose, CA, USA). The calculated cytotoxicity percentages were adjusted by deducting the values from blank wells (without cells). For the Cur-ND assay, U-87 MG cells were treated with either 0, 5, 10, 15 µM of free curcumin or Cur-ND for durations of 24 or 36 h, using the same methodology as described.

### 2.8. Cellular Uptake of APA NDs

To investigate the cellular uptake of various APA NDs, human cell lines U-87 MG and HepG2 were initially seeded at a density of 1.5 × 10^5^ cells per well in a 6-well plate and cultured in 5% CO_2_ at 37 °C for 24 h before the commencement of ND treatment. The cells were then treated with concentrations of 0, 0.05, 0.10, 0.15, 0.20, 0.25 mM of different ND formulations, including [AA]_2_, [AB]_2_, [BB]_2_, [AC]_2_, [CC]_2_, [AA]_3_, [AB]_3_, [BB]_3_, [AC]_3_ and [CC]_3_ NDs for a period of 12 h. The uptake of these NDs by the cells was quantified using CytoFLEX Flow Cytometer (Beckman Coulter Ltd., Brea, CA, USA), equipped with a FITC filter set.

### 2.9. Molecular Dynamics Simulations of APA NDs

ND template preparation for MD simulations. For the creation of ND systems, membrane scaffold protein templates from CHARMM-GUI were utilized. To match the ND sizes accurately, protein templates with lengths comparable to the total lengths of APAs (with varying single-stranded chain lengths) were selected. For APAs with chain lengths of 22 or 66, MSP1 was employed, while MSP1Δ1-22 was used for APAs with a chain length of 44. DMPC was integrated with these protein templates to form two control NDs using CHARMM-GUI, maintaining a 1:1 ratio of DMPC in the upper and lower leaflets.

Construction of APA NDs. Fifteen homologous models of APAs were developed using MOE 2019. From these, the model showcasing the lowest root-mean-square deviation (RMSD) from the average structure was chosen for MD simulation. The APA structures predominantly comprised α-helixes and matched the protein templates perfectly. These templates were then removed to yield NDs composed of APA and DMPC. Energy minimization was performed to resolve any atomic clashes within the ND.

Simulation system preparation. NDs were placed in a cubic box, ensuring a minimum distance of 1.2 nm from the boundary. The ND systems were solvated using the TIP3P water model and neutralized with 0.145 M NaCl. The CHARMM36 force field was applied for all-atom MD simulations. Systems underwent a maximum of 5000 steps of minimization using the steepest descent algorithm.

Simulation Parameters Setup. An NVT equilibration for 0.25 ns with a 1 fs step size was followed by an NPT balance of 1.625 ns, initially with a 1 fs time step for 0.125 ns, then 2 fs for the remaining 1.5 ns. The Berendsen algorithm maintained the temperature at 298 K during the NVT and NPT stages and the pressure at 1 bar during the NPT stage. NDs were initially position-restrained, gradually reducing the intensity until complete release by the end of the equilibration stage. Each system underwent a 100 ns simulation with a 2 fs time step. Temperature and pressure were maintained at 298 K and 1 bar, respectively, using the Nosé–Hoover and Parrinello-Rahman algorithms. The LINCS algorithm constrained bonds containing hydrogen. Long-range Coulomb interactions were calculated using particle-mesh Ewald, with a van der Waals cutoff radius of 1.2 nm and a rvdw-switch set at 1.0 nm. GROMACS 2020.5 conducted all simulations, and trajectory analyses were performed using GROMACS and VMD 1.9.4.

Trajectory Analysis. In trajectory analysis, intervals between frames were set at 100 ps, utilizing all 1000 frames in the simulation. The RMSDs of DMPC, APAs, and the nanodisc complex were calculated accordingly.

### 2.10. APA Structure Prediction Using ColabFold

The structure prediction for each of the five monomers of APA variants was conducted independently using the ColabFold-AF2 web-based prediction tool [36]. The process begins with opening a web browser and navigating to the ColabFold website at https://alphafold.colabfold.com (accessed on 23 January 2023). By default, the GPU is enabled for use. The amino acid (AA) sequence of APA is then entered into the ‘query_sequence’ field on the site. To initiate the prediction process, select ‘Runtime’ from the menu bar and click on ‘Run all’. ColabFold is set to automatically generate five structural models for each query. As each model is completed, the predicted structures and their corresponding results are shown on the website. These can then be downloaded for further analysis.

### 2.11. Statistical Analysis

Results are expressed as mean ± standard deviation (SD) unless otherwise noted. Two-way ANOVA and Student’s *t*-test were conducted using GraphPad Prism10 software to determine the significance between the groups. A *p*-value of less than 0.05 was considered to be statistically significant (* *p* < 0.05, ** *p* < 0.01, *** *p* < 0.001, **** *p* < 0.0001).

## 3. Results and Discussion

### 3.1. Design and Structure Modeling of apoA-I Peptide Analogs

Three apoA-I α-11/3 helical mimics, identified as Chain A, B, and C, were initially designed as foundational elements for the construction of fifteen distinct APAs. These APAs were differentiated based on their combinatorial arrangement and oligomerization states, presenting in the form of tandem repeats comprising 22, 44, or 66 AAs (Figure 1A). This design strategy was grounded in a previously established consensus-based sequence normalization approach [33]. A critical aspect of our research involved the strategic insertion of specific AAs near the C-terminal end of these chains. This modification is aimed at facilitating and monitoring morphological changes in the NDs assembled from these APAs. The overarching goal of these investigations is to identify APA sequences with the most promise for application in ND-mediated drug delivery based on their size, stability, and morphological properties.

We predicted structures for a series of APA sequences using ColabFold-AF2. These analogs are identified by two-letter acronyms that represent variations in their AA sequences, specifically differing in their content of Serine (S/Ser), Proline (P/Pro), and Tryptophan (W/Trp). The accuracy of these structural predictions was assessed using two key metrics: the predicted Local Distance Difference Test (pLDDT) and the predicted Template Modeling score (pTM). The pLDDT scores, averaged across all residues of an APA chain, indicate the overall confidence in the predicted structure of the entire chain. As depicted in Figure 1B, these pLDDT values are shown in a blue gradient and range from 81.6 to 97.4. This range highlights a significant variation in the confidence levels associated with the local structural elements of these analogs. Notably, all obtained pLDDT scores exceeded 80, suggesting a high level of reliability in the predicted tertiary structures of the APA peptides. The [AA]_3_ analog exhibited the highest confidence in its structure, as indicated by a pLDDT score of 97.3, suggesting a robust and reliable prediction of its tertiary architecture. In contrast, the [CC]_3_ analog showed the lowest confidence score at 81.6. This lower score may indicate a potential intrinsic disorder or a more dynamic nature in this particular peptide, signaling less certainty in its local structural prediction.

Conversely, the pTM scores, presented in a green gradient, provide a measure of the accuracy of the overall fold of the peptides. These scores, which range from 0.18 to 0.69, are critical in evaluating how closely the predicted global structure of the peptides aligns with their actual physical forms. The pTM scale, spanning from 0 to 1, allows AF2 to prioritize its predictions. Scores below 0.2 typically signify randomly arranged residues with little to no resemblance to the native structure, or they may indicate intrinsically disordered proteins. On the other hand, a pTM score exceeding 0.5 is generally considered indicative of a reliable structure prediction. Among the analogs, the [AA]_3_ variant stands out with a pTM of 0.69, suggesting that its predicted structure is likely a close approximation of its native state in physiological environments. The [CC] analog, with the lowest pTM score of 0.18, points to considerable uncertainty in its predicted structure, possibly due to a lack of similar structures in the training dataset or inherent flexibility within the peptide.

A notable trend is observed in APA analogs characterized by tandem repeated units such as [AA]_2_, [AB]_2_, [BB]_2_, [AA]_3_, [AB]_3_, and [BB]_3_, which generally exhibit higher pTM scores. This could imply that repetitive sequences within these analogs contribute to a more predictable and stable overall structure, potentially through the formation of repetitive secondary structural motifs like alpha-helices, leading to more defined tertiary structures. The study reveals that when APA analogs incorporate chains A or B, both pLDDT and pTM scores yield the desired results. However, the inclusion of chain C introduces greater uncertainty in structure prediction, particularly when two chain Cs are present. This suggests that the addition of Serine and Tryptophan near the C-terminal of chain C elevates the uncertainty of the structure prediction.

The best model output from AF2 for each analog is visually represented in Figure 1C. Here, the cartoon structures of the analogs are color-coded to reflect per-residue confidence levels, as determined by the pLDDT scores. The color gradient, ranging from blue to yellow, indicates varying degrees of confidence in local structure predictions. Notably, the AA and AB analogs, in both their monomeric and oligomeric forms, predominantly exhibit a blue coloration, signaling high pLDDT scores and, hence, a strong confidence in their local structural predictions. This uniformity in color suggests that AA and AB analogs maintain a stable and well-defined conformation across various oligomeric states. In contrast, the AC and CC analogs are marked by a significant presence of green and yellow hues, particularly in their trimeric forms ([AC]_3_ and [CC]_3_), denoting lower pLDDT scores. This variation indicates reduced confidence in the local structural predictions for these analogs and may point to an inherent flexibility or disorder, potentially amplified by oligomerization. The BB analogs display an intermediate pattern with a mix of blue and light blue regions, suggesting moderately high level of confidence in their structure. The increase in light blue regions in their trimeric forms compared to the monomeric and dimeric states could imply diminishing confidence in structure prediction with increasing oligomerization, indicating potentially less certain conformations in higher-order oligomeric states.

Additionally, the influence of Proline content on structural predictions is noteworthy, especially considering its role in introducing kinks and flexibility, which can shape NDs. A higher Pro content in APA analogs suggests a greater likelihood of accumulative α-helical conformational bending and twisting, potentially beneficial for creating tunable ND formations. Overall, the AF2-derived results unveil a complex interplay between the peptide sequence of APA analogs, their oligomerization, and structural stability. These findings could be instrumental for future experimental validation of the predicted structures and in understanding the functional consequences of the observed conformational tendencies. Furthermore, this modeling data is crucial in the design of APA-based NDs, where precise prediction and manipulation of ND conformations are fundamental to achieving specific therapeutic outcomes. Accurate structural predictions enable the fine-tuning of ND properties, including stability, solubility, and drug loading capacity, all of which are vital for effective drug delivery. The capability to adjust ND conformations as per predictive models allows for the customization of these NDs for various medical applications, particularly in targeted drug delivery and enhancing bioavailability. The integration of this modeling data into the development process represents a pivotal advancement in the field of nanomedicine, offering a more precise and targeted approach to disease treatment and management.

### 3.2. Molecular Dynamics Simulations of APA-ND Systems

To explore the dynamic behavior of NDs when assembled with various APA analogs, we performed a total of 1.7 μs long (100 ns each) all-atom MD simulation, spanning 100 ns for each of the 17 distinct ND systems. Our primary objective was to discern the impact of APAs on the dynamic stability of these ND assemblies in comparison to established scaffold proteins, namely MSP1 and MSP1Δ1-22. We meticulously analyzed both the mean root mean square deviation (µRMSD) of the ND systems and the µRMSD of the APA analogs and scaffold proteins over the course of these simulations, as visually represented in Figure 2A,B. The ND systems were categorized based on their lipid-to-APA ratios, which included 188:2, 170:2, 188:18, 170:8, and 188:6. These ratios were reflective of the varying concentrations of APAs, contingent upon their sequence lengths within the NDs. We employed ND systems assembled with either MSP1 or MSP1Δ1-22 as a reference framework for our comparative analysis. The results show that the ND_µRMSD values fluctuated between approximately 9 and 14 Å. Intriguingly, monomeric APAs such as AA, AB, AC, BB, and CC demonstrated stability akin to the MSP1 scaffold, with an exception for those possessing a higher content of Proline (Pro) residues. However, a distinct pattern emerged with the oligomeric APAs, including [AA]_2_, [AB]_2_, [AC]_2_, [BB]_2_, [CC]_2_, [AA]_3_, [AB]_3_, [AC]_3_, [BB]_3_, and [CC]_3_. These displayed notably lower ND_µRMSD values, particularly pronounced in the AA and AB oligomeric APA-NDs. This pattern led us to suggest that increased adoption of Proline, Serine, and Tryptophan within these analogs may be a contributing factor to the observed variability in ND stability.

In examining the intrinsic dynamic stability of APA analogs and scaffold proteins within ND assemblies, the Protein_µRMSD is calculated as a crucial metric, as depicted in Figure 2B. For most APA analogs, except for the trimeric forms [AA]_3_ and [AB]_3_, µRMSD values were observed to be higher than those of their reference scaffold proteins. This disparity can be partially attributed to the scaffold proteins having fewer chain segments and splitting gaps compared to the APAs, with their sufficient length likely contributing to enhanced dynamic stability. The APAs displayed a spectrum of stability, with trimeric forms incorporating limited Proline ([AA]_3_, [AB]_3_, [AC]_3_) generally showing lower µRMSD values, indicative of stability comparable to the MSP1 scaffold in their interaction with the lipid bilayer. Interestingly, certain oligomeric APA forms exhibited increased µRMSD, potentially reflecting fewer stable conformations within the ND, possibly due to altered lipid-residue interactions in oligomeric units. The variation in µRMSD across different lipid-to-APA ratios and oligomerization states underscores the complexity of interactions within ND systems. The findings suggest that the concentration and oligomeric state of APAs, as well as specific residue incorporation, significantly impact ND stability. Further insights were gained by quantitatively measuring the APA-mediated hydrogen bond (H-bond) frequency, as shown in Figure 2C. A higher frequency of H-bonds, generally associated with increased structural stability, was observed for all APA analogs compared to the reference scaffold proteins, with frequencies ranging approximately from 60 to 100 H-bonds per 100 picoseconds. This indicates that individual APAs can establish stable interactions within the ND, potentially through H-bonds with lipid molecules or within the APA intermolecular H-bond network. The MSP1 scaffold protein exhibited the lowest H-bond frequency in its standard form, with a slight increase following N-terminal truncation (MSP1Δ1-22), implying that this truncation does not significantly impair, and may even enhance, hydrogen bonding capabilities. Notably, APAs incorporating chain C showed a decreased H-bond frequency, likely due to the replacement of Arginine and Lysine with Serine and Tryptophan. Remarkably, oligomeric BB analogs demonstrated a higher H-bond frequency compared to AA analogs, suggesting that BB oligomers not only maintain their H-bonding capabilities but may also offer additional bonding opportunities through intermolecular interactions between APA units.

The solvent-accessible surface area (SASA) data offers vital insights into the dynamic conformational states of proteins within ND systems. For monomeric APA-based ND assemblies, SASA values demonstrate minimal variation across different APA types, indicating a consistent exposure to the solvent regardless of the specific ND system. This consistency suggests that monomeric APAs maintain a relatively stable conformation within the ND environment. In the context of oligomeric APAs with a higher Proline (Pro) content, there is a noticeable trend: SASA values tend to decrease as oligomerization increases. This observation might indicate that oligomers with increased Pro incorporation adopt a more compact structure, leading to reduced solvent exposure. Such compaction could result from the conformational alterations induced by the Proline kink effect within the APA units in the ND system. Interestingly, a greater variation in SASA is observed in Proline-free APAs, such as the AA analogs, suggesting a more rigid conformational structure for these ND assemblies. Furthermore, the scaffold proteins MSP1 and its truncated variant MSP1Δ1-22 display distinct SASA values. The truncated MSP1Δ1-22 generally exhibits a lower SASA compared to the full-length MSP1, which can be attributed to the removal of the N-terminal region, resulting in a smaller ND system and thus a reduced SASA. These SASA measurements imply that both the composition and oligomeric state of APAs significantly influence the conformational dynamics and stability of ND assemblies. Understanding these dynamic properties is essential for the functional characterization of NDs, particularly in their potential use in therapeutic delivery systems. These findings particularly highlight that AB oligomers ([AB]_2_ and [AB]_3_) exhibit superior stability compared to other APA types and their reference scaffold proteins, underscoring their potential significance in ND-based therapeutic applications.

Next, we delve into a comprehensive analysis of the effects of proline and tryptophan residues on the conformational changes of specific APA-NDs. By examining the central structures within the principal conformation clusters of APA-NDs from 100 ns MD simulations, we obtained the core structures for reference NDs with scaffold proteins MSP1 and MSP1D1-22, as depicted in Figure 3A. These final modeled structures of various APA-ND types are crucial for assessing the impact of distinct design parameters on their morphological alterations. Although these ND models started with fixed initial structures, the APA sequences, diverse in nature, instigated alterations in the ND surface curvature (NDSC), which is intimately linked with the morphological behavior of the APA-NDs. We observed that the deformation of DMPC bilayers is a key factor influencing the NDSC. To quantify this effect, we measured the geometric mean inscribed angle ($\overset{¯}{\theta }$) of the NDs, as shown in Figure 3B. Considering the irregular shapes of ND structures, $\overset{¯}{\theta }$ was deemed an appropriate metric to encapsulate the entire range of NDSC variations, thereby providing a holistic understanding of how proline and tryptophan residues influence the dynamic morphology of APA-NDs within the simulation environment.

In the cases of AA, AB, and BB series APA-NDs with varying chain lengths (22, 44, and 66 AAs), it can be observed that the increased substitution of chain B for chain A in these NDs led to a reduction in the $\overset{¯}{\theta }$ (Figure 3C), suggesting increased curvature formation with twisted conformation in proline-rich AB and BB NDs (Figure 3D) [37,38,39]. Conversely, in the AC and CC series NDs, where there is an increase in Serine-Proline-Tryptophan (Ser-Pro-Trp) substitution in chain C, an opposite trend in $\overset{¯}{\theta }$ was observed. These two contrasting behaviors in NDSC appear to closely align with the µRMSD results. We hypothesize that the presence of tryptophan following a proline residue could introduce steric hindrance, impeding the proline-induced kink typically seen in α-helical APAs. This interaction between tryptophan and proline may lead to a decrease in bending, consequently resulting in flatter NDs. Additionally, the oligomeric state of the APAs seems to influence NDSC variations. NDs formed by monomeric APAs exhibited more splitting gaps, indicating a tendency to cause local dislocation or disorganization of phospholipids at the deformation termini of the APAs. This results in significant variations in $\overset{¯}{\theta }$, suggesting that NDs assembled with monomeric APAs may exhibit unexpected geometric curvature characteristics, distinct from those induced by proline kinks. From these analyses, we provide evidence that an increase in proline content within specific APAs can modulate the geometric properties of NDSC. However, it is important to note that the morphological behavior of APA-NDs is not solely influenced by site-specific proline substitution. Rather, it is determined by a combination of factors, including the incorporation of amphiphilic residues like tryptophan and the oligomeric state of the APAs. This multifaceted interaction underscores the complexity of the structural dynamics governing APA-NDs.

### 3.3. Hydrodynamic Diameter Measurements of ND Systems

To further elucidate the particle size and temporal stability of APA-NDs, we synthesized a series of NDs constituted by DMPC and oligomeric APAs. Our previous studies found that an increase in the DMPC-to-APA ratio results in a larger ND diameter and impacts the uniformity of the ND perimeter [33]. Herein, a specific DMPC-to-APA ratio of 15:1 has been selected for the construction of NDs and to optimize the long-term stability of ND suspensions. Additionally, the solubilizing influence of sodium cholate (NaCh) on phospholipid molecules and its role in restricting membrane fluidity are critical in supporting short APAs to sustain stable ND assemblies with smaller diameters, thereby adding a layer of complexity to the formulation design [33]. In this context, NDs constituted from 44- and 66-mer APAs at a DMPC-to-APA ratio of 15:1 have been chosen for detailed long-term stability assessments. The storage stability of these DMPC NDs, stabilized by APAs, was evaluated over a span of 8 weeks using dynamic light scattering (DLS), as illustrated in Figure 4. When stored at 4 °C, the NDs exhibited APA sequence-dependent stability. NDs with [AA]_2_ and [AA]_3_ APAs demonstrated remarkable stability, with only approximately a 1–2 nm alteration in hydrodynamic diameter (dH) over the course of 8 weeks. NDs comprising Proline-containing APAs, such as AB and BB, manifested smaller particle sizes and displayed comparable stability, with ND dH variations within an approximate range of 1–3 nm. In contrast, APA-NDs with sequences incorporating both Proline and Tryptophan (AC and CC) revealed the least stability and displayed a marked increase in dH at the eight-week mark, hinting at potential aggregation or structural reorganization over the period. Intriguingly, we found that higher levels of proline incorporation tended to result in the formation of smaller NDs. Conversely, NDs formed from APAs containing chain C motifs demonstrated an increase in dH, suggesting an adverse effect of the Ser-Pro-Trp mutation on the formation of smaller NDs. The [AA]_2_, [AA]_3_, [AB]_2_, and [AB]_3_ APA-ND systems generally exhibited little to moderate change in particle size over time, signifying that these particles maintained stable colloidal properties without significant aggregation or dissociation. These empirical observations align well with the theoretical interpretations derived from MD simulation assessments. Consequently, it appears that ND systems derived from AA and AB oligomeric APAs may be more desirable for applications where long-term colloidal stability is requisite.

### 3.4. Dynamic Behavior of Curcumin-Loaded APA-NDs Formulation

We investigated the dynamic properties and stability of [AA]_3_/DMPC ND formulations encapsulating curcumin through 100 ns all-atom MD simulations. Curcumin was incorporated into the NDs in three different proportions: 5% to 10%, 21%, and 42% of the total lipids, each randomly positioned on the ND membrane bilayer surface. The structural stability of the ND and the MD of the curcumin within the DMPC membrane were analyzed by calculating the RMSD of the DMPC lipids, the MSP1/APA scaffold proteins, the complete ND complex, and the curcumin molecules over time, as detailed in Figure 5A–E. For the bare ND system without curcumin (Figure 5A), MSP1 achieved equilibrium within the first 80 ns, with DMPC and MSP1 showing parallel trends and an average RMSD of 1.3 nm and 1.0 nm, respectively, after 50 ns. The global motions of the ND complex in the absence of curcumin peaked at an RMSD of approximately 1.3 nm after 90 ns. In stark contrast, the ND assembly with a 5% curcumin load exhibited instability in the initial 50 ns (Figure 5B), with RMSD values progressively rising to around 1.3 nm towards the end of the simulation. Notably, the average RMSD during the latter half of the simulation for DMPC and [AA]_3_ remained at 1.3 nm and 0.9 nm, respectively, indicating that [AA]_3_ retained stability akin to MSP1 in the curcumin-free ND system, seemingly unaffected by the presence of curcumin molecules.

To further study the impact of curcumin encapsulation on the stability of NDs, we conducted simulations with NDs containing elevated curcumin contents of 10%, 21%, and 42% of the total lipids. It was observed that the structure of the ND complex with 10% curcumin became increasingly stable, with the highest RMSD exceeding 1.1 nm. Yet, the average RMSD values for DMPC and [AB]₃ in the latter half of these simulations were 1.2 nm and 0.7 nm, respectively, which interestingly shows a reduction in RMSD compared to the 5% curcumin ND system (Figure 5C). This reduction suggests that at higher concentrations, curcumin may aid in stabilizing the DMPC and APA molecules within the ND complex. For NDs with higher curcumin contents of 21% and 42%, the RMSD values generally increased, reaching a maximum between 1.0 and 1.1 nm (Figure 5D,E). However, the average RMSD for [AA]_3_ remained constant at 0.7 nm across these higher concentrations, while the average RMSD of DMPCs fluctuated between 1.1 and 1.2 nm, indicating a plateau in stabilization provided by the increasing amounts of curcumin.

This trend implies that the beneficial effect of curcumin in restraining the mobility of DMPC and APAs occurs when the curcumin content is between 10% and 21%. A higher proportion of curcumin seems to enhance the overall stability, as evidenced by the reduced RMSD values of APA molecules. Our simulations indicate that curcumin is likely to form stable interactions with the APA and DMPC components. Drug leakage assessments revealed that between 20 and 50% of curcumin dissociated from the NDs over the course of 100 ns MD simulations, with this dissociation being concentration-dependent (Figure 5G). Moreover, higher concentrations of curcumin were found to self-aggregate, driven by hydrophobic interactions between the curcumin molecules (Figure 5F,G). Should a curcumin-to-DMPC ratio of 10% be chosen for ND formulation, the drug EE% and drug LC% are anticipated to be 65% and 2.7%, respectively, demonstrating the trade-off between drug loading and structural stability in ND systems.

Furthermore, an analysis of the molecular interactions between curcumin and the components of the APA-ND system was conducted. As illustrated in Figure 5H, electrostatic interactions were observed, particularly salt bridges and H-bonds, forming between the functional groups of curcumin, specifically the ketone and hydroxy groups, and the lysine and arginine residues within the APA peptides. Additionally, the DMPC molecules were found to engage with the hydroxy groups of curcumin through hydrogen bonding and with its phenyl rings via weak electrostatic and dispersion force interactions. These interactions are critical as they suggest a strong integrative relationship between curcumin and both APA peptides and lipid components of the ND, providing a molecular basis for the observed stability and encapsulation characteristics within the system.

### 3.5. Tunable APA-NDs Enable Customized Screening for ND Drug Formulation

Tunable APA-NDs are being leveraged for the development of ND drug formulations. In this regard, several critical factors are considered, including long-term stability, cellular uptake efficacy, biosafety, and ND size. Next, we employed flow cytometry (FCM) to assess the cellular uptake of DiOC14-labeled APA-NDs. Upon 12 h of exposure, Hep G2 and U-87 MG cells displayed a marked concentration-dependent escalation in APA-ND internalization, as evidenced by the increased fluorescence intensity of FITC (Figure 6A). Particularly, cells treated with [AA]_3_ and [CC]_3_ NDs showed substantially higher uptake than those treated with NDs of other APA compositions. Note that [CC]_3_ NDs showed poor size stability and uniformity compared to their counterparts. In light of the observed stability and endocytosis efficiency, 44-mer and 66-mer [AA] APAs were chosen for further cytotoxicity testing using the MTT assay in Hep G2, U-87 MG, and NCM460 cell lines. These cells, in their log growth phase and thus more sensitive to external agents, revealed that their viability remained unaltered by the administration of varying lengths of [AA] APAs at concentrations from 0 μM to 1 μM. This indicates a high level of cell biocompatibility for [AA] APAs derived from apoA-I across both tumor and normal cell lines (Figure 6B). Such findings propose that the [AA]_2_ and [AA]_3_ NDs could constitute a viable platform for drug formulation, combining effective cellular delivery with biocompatibility.

Therefore, we selected [AA]_3_ as a representative APA to evaluate the Cur ND formulation strategy. We prepared an APA-ND formulation using DMPC and [AA]_3_ APA at a reconstitution ratio of 15:1 to encapsulate Cur. The formulation achieved an encapsulation efficiency of 78.1% and a drug loading capacity of 2.8%. The antiproliferative activity of this Cur-encapsulated ND was assessed against U-87 MG cells, with treatments including vehicle control, free Cur dissolved in DMSO, and Cur encapsulated in Cur-NDs, with cell viability measured at 24 and 36 h (Figure 6C). The Cur-NDs significantly outperformed free Cur in inhibiting cell proliferation, with a statistically significant difference in cell viability observed at both time points and across all three tested concentrations (*p* < 0.05). These results illustrate that ND drug formulations with optimized properties can be realized through the careful design and selection of APA sequences.

## 4. Conclusions

The findings from our current research resonate with previous insights while broadening our understanding of ND formulation dynamics. We observed that the design and structure of APAs play a pivotal role in ND formation and stability. Our predictive models, using predictive tools like ColabFold-AF2, revealed significant correlations between APA sequence variations, especially the inclusion of Proline, Serine, and Tryptophan, and the structural integrity of NDs. This echoes our prior understanding of the critical role of AA composition in determining ND properties. Furthermore, the MD simulations conducted on various APA-ND systems highlighted the influence of APA oligomeric states on ND stability. These simulations revealed interesting patterns in the behavior of NDs when complexed with different APA variants. For instance, oligomeric APAs demonstrated distinct stabilities and interactions compared to their monomeric counterparts. This reinforces the concept of tunability in ND design, where adjusting the oligomeric state and sequence of APAs can tailor ND properties for specific applications. Moreover, the long-term stability assessment of NDs composed of oligomeric APAs offered practical insights into their potential for real-world applications. Particularly noteworthy was the stability of NDs comprising chain A and/or chain B APAs, which suggests a promising avenue for the development of stable and effective ND-based drug delivery systems. Our foray into the dynamic properties of curcumin-loaded APA-NDs further exemplified the versatility of these constructs. The stability and interaction dynamics of these formulations, analyzed through MD simulations, underscored the potential of APA-NDs in encapsulating and effectively delivering bioactive compounds like curcumin.

In conclusion, our study enriches the existing knowledge pool on NDs and APAs, offering valuable insights into the rational design of these systems for biomedical applications. The intricate interplay between APA sequence, structure, and ND stability highlighted in our research paves the way for the development of customizable NDs for targeted drug delivery. Looking forward, further exploration into the interaction dynamics of different bioactive compounds with APA-NDs and their in vivo behavior will be crucial in realizing the full potential of these systems in therapeutic contexts. This research thus not only contributes significantly to the field of nanomedicine but also sets a foundation for future innovations in drug delivery systems.

## Figures and Tables

**Figure 1 bioengineering-11-00245-f001:**
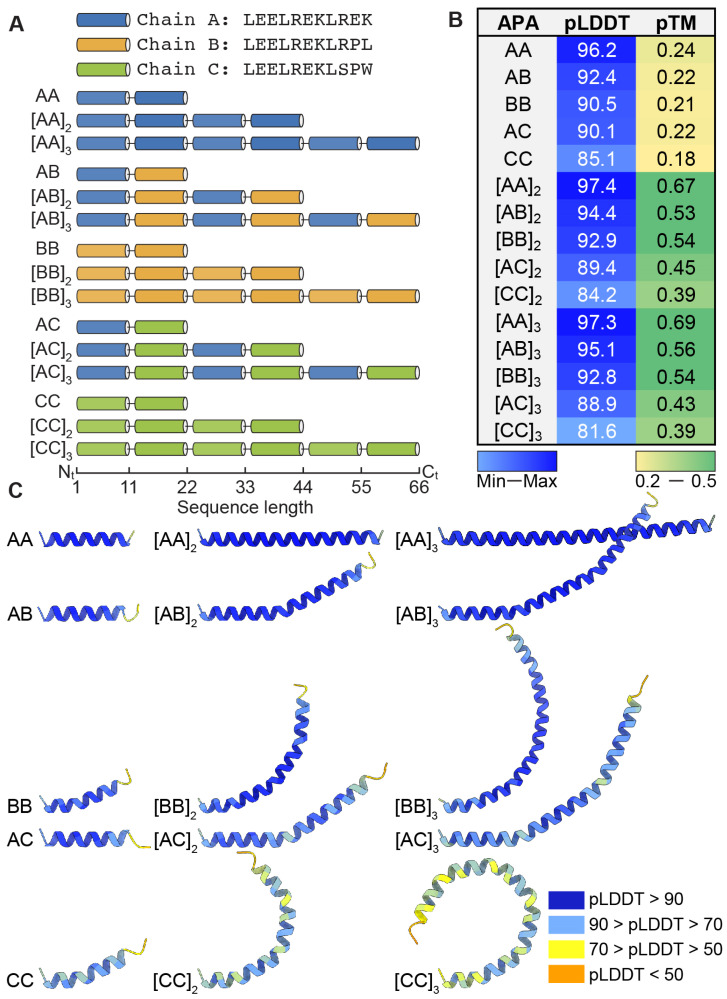
Structural predictive quality assessment of APAs. (**A**) the illustration of the primary sequences of APAs, categorized by chain type and length. The varying sequence lengths, from monomers to trimeric repeats, are illustrated to convey the incremental increase in the number of AAs as the oligomerization degree progresses. (**B**) The quality of structural predictions for each APA is quantified by the pLDDT and the pTM scores. These metrics provide insights into the confidence and accuracy of the predicted tertiary structures of APAs, ranging from monomers to trimers. (**C**) The predicted tertiary structures of APAs, with color-coding based on the pLDDT score. The colors range from blue, indicating high confidence in the local structure prediction, to yellow, signifying lower confidence. The illustrations demonstrate the helical configurations of the APAs, with variations in the degree of helical bending and kinking, particularly in relation to the presence of proline and tryptophan residues.

**Figure 2 bioengineering-11-00245-f002:**
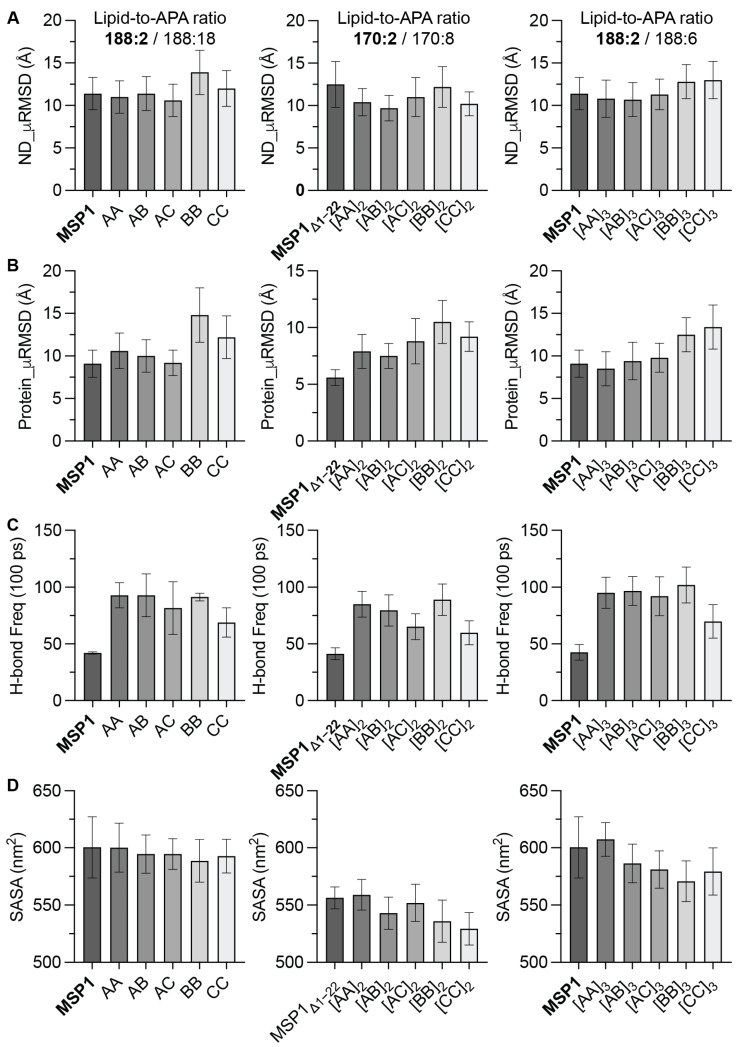
Comparative analysis of stability and interactions of APA-NDs across two lipid-to-APA ratios and different compositions. (**A**) The comparison of the ND_µRMSD values for various APAs and oligomerization states, including the reference scaffold proteins MSP1 and MSP1Δ1-22. The results show differences in stability among the different APA-ND systems. (**B**) The protein_µRMSD for each APA type within the ND systems indicates the intrinsic dynamic stability of the proteins. (**C**) The frequency of H-bond formation within the ND systems is indicative of interaction strength and structural integrity between the proteins and the lipid bilayer. (**D**) The comparison of SASA values among the various APA types and their oligomeric states within ND systems indicates how the structure and composition of APAs may influence the overall conformation and interaction of the NDs with their environment.

**Figure 3 bioengineering-11-00245-f003:**
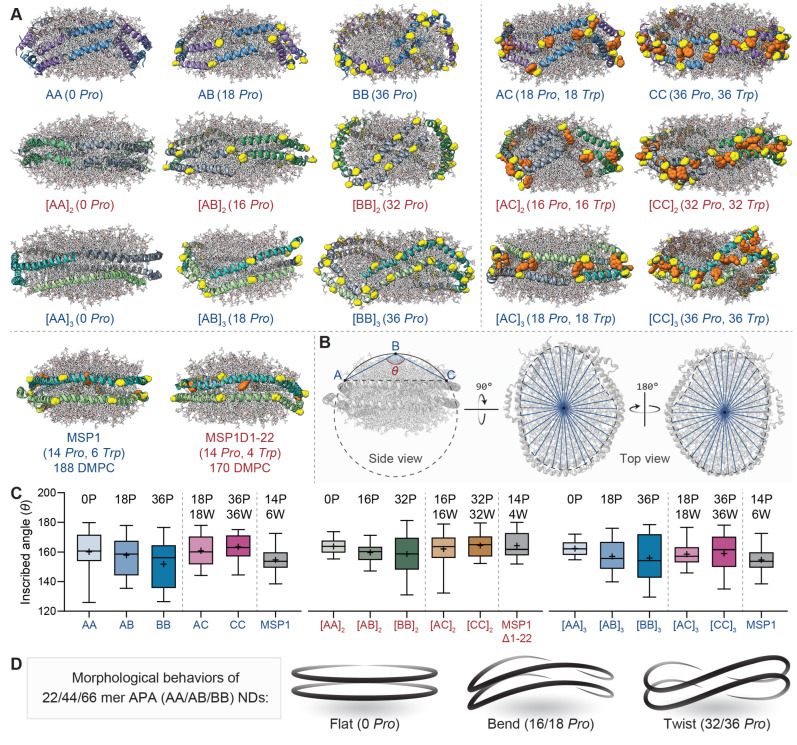
The morphological correlates of sequence changes in APA-ND central structures. (**A**) Illustration of central structures of fifteen APA-NDs and two reference MSP-NDs. Here, the central structures demonstrated the representative conformation with the minimum average RMSD from the main APA-ND clusters among 100 ns MD simulations. For 22- and 66-mer APA-NDs, the central models were generated from the template MSP1 (188 DMPC) surrounded by 18 and 6 copies of APAs, respectively. 44-mer APA ND models were adapted from the truncated template MSP1Δ1-22 (170 DMPC) and surrounded by eight copies of APAs. For each central structure, adjacent chains are displayed in different colors, the proline residues are colored in yellow and the tryptophan residues are colored in orange. (**B**) The instruction of the geometric mean inscribed angle ($\overset{¯}{\theta }$) of NDs. Each $\overset{¯}{\theta }$ was accessed by setting two chords that intersect with the geometric center of the ND plane (the phosphorus atom was selected as the measurement point), and the angle ABC was measured as an inscribed angle. (**C**) The box plots show the mean inscribed angle ($\overset{¯}{\theta }$) of different APA-NDs and their reference structures. (**D**) The schematic description for the morphological change in series [AA], [AB], and [BB] APA-NDs. Which undergoes morphological deformation due to the accumulative distortion produced by increased proline content.

**Figure 4 bioengineering-11-00245-f004:**
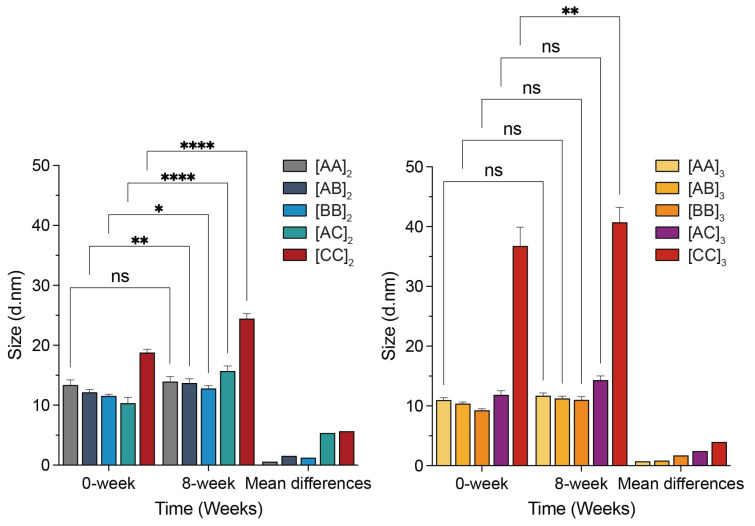
Temporal analysis of hydrodynamic diameter (dH) variations in 44- and 66-mer APA-NDs systems over eight weeks when stored at 4 °C, providing insight into the stability and aggregation behavior of these ND systems. Values are means ± SD (*n* = 4). Statistical evaluation was conducted using GraphPad Prism10, employing Two-way ANOVA for multiple comparisons. The significance levels are denoted as follows: n.s. indicates a lack of statistical significance; * *p* < 0.05; ** *p* < 0.01; and **** *p* < 0.0001, reflecting varying degrees of statistical relevance.

**Figure 5 bioengineering-11-00245-f005:**
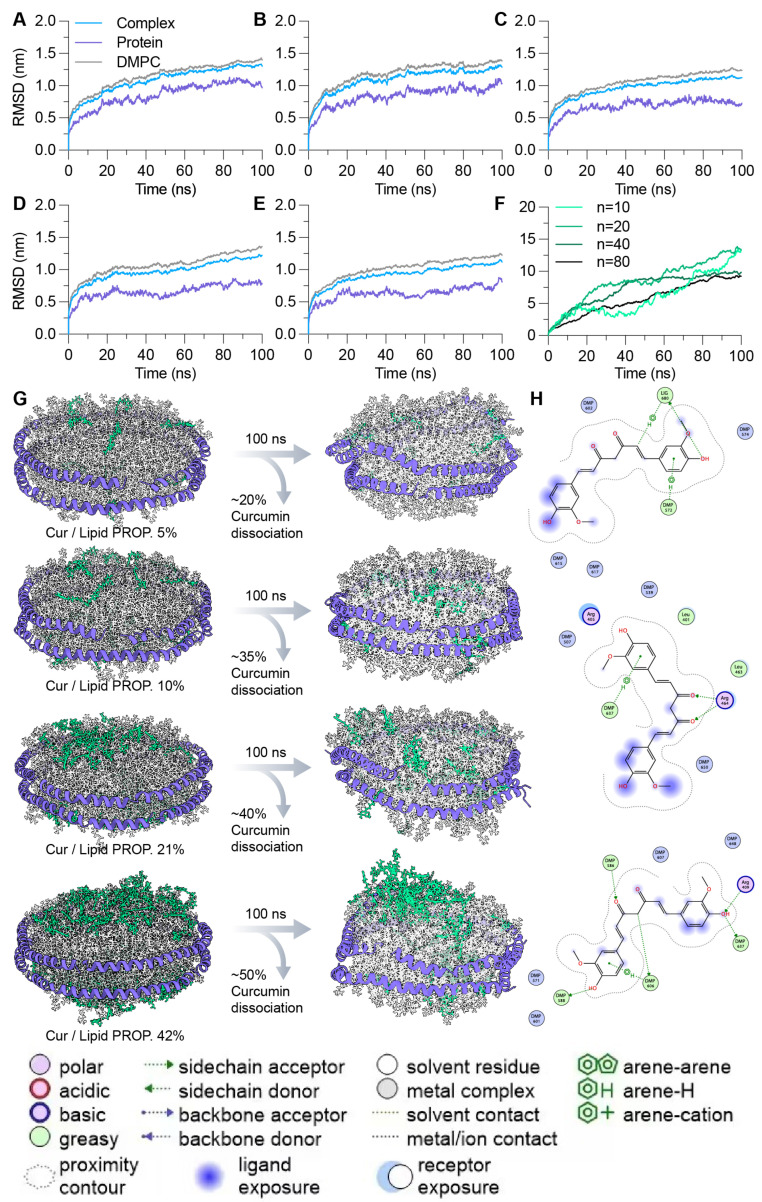
Stability and interaction dynamics of curcumin-encapsulated APA-NDs at varied concentrations. (**A**–**E**) RMSD trajectories over 100 ns for the ND complex, protein, and DMPC components, depicting the impact of varying curcumin loads on ND stability. (**F**) The RMSD of curcumin molecules themselves, detailing the stability of curcumin within the ND system at different lipid proportions. (**G**) MD-based drug leakage assessments, illustrating the percentage of curcumin dissociation from the NDs at different curcumin-to-lipid ratios over 100 ns simulations. (**H**) The molecular interactions between curcumin and APA-ND components, focusing on key electrostatic interactions and H-bonds that contribute to the stability and integrity of the ND system.

**Figure 6 bioengineering-11-00245-f006:**
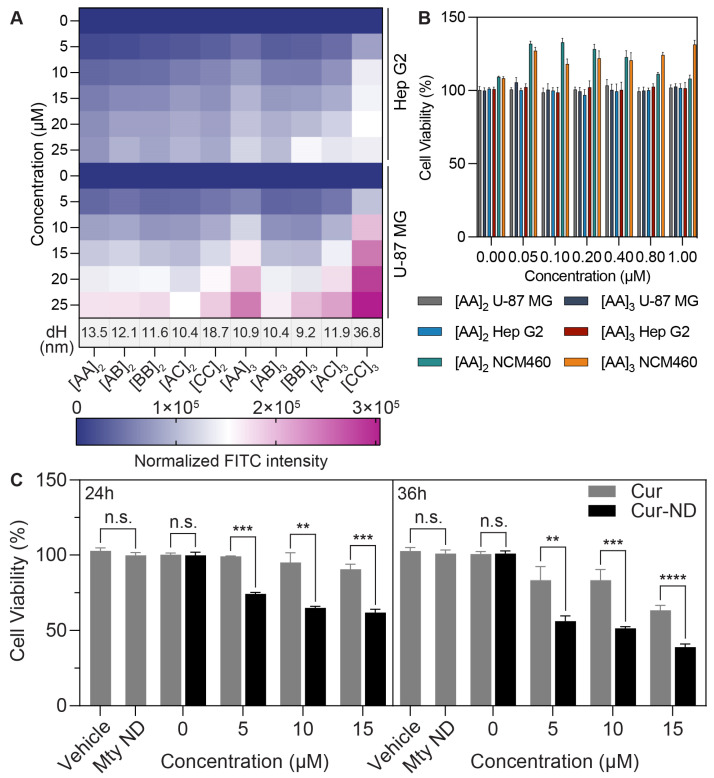
Tunable APA-NDs enable customized screening for ND curcumin (Cur) formulation. (**A**) The heat map for the effect of APA-NDs concentration/dH on Hep G2 and U-87 MG uptake efficacy. (**B**) The histogram for the effect of APA-NDs ([AA]_2_ and [AA]_3_) on the viability of tumor cells Hep G2 and U-87 MG and normal cell NCM460. Cells were incubated with increasing APA concentration (0 to 1000 μg/mL) for 24 h. (**C**) The comparative evaluation of the Cur anti-proliferative activity between free drug form and [AA]_3_ Cur-ND formulation. Cells were treated with specified concentrations (0, 5, 10, 15 μM) of free Cur (in DMSO) or Cur-NDs for 24 and 36 h. Values are means ± SD (*n* = 4). Differences between groups for the 24 and 36 h time points are shown in the figure. n.s., not significant; ** *p* < 0.01; *** *p* < 0.001; **** *p* < 0.0001.

## Data Availability

The raw data supporting the conclusions of this article will be made available by the authors.

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
