# Peer review of "Molecular Dynamics and In Vitro Studies Elucidating the Tunable Features of Reconfigurable Nanodiscs for Guiding the Optimal Design of Curcumin Formulation"

_bioengineering, 2024, doi:10.3390/bioengineering11030245_

Round 1

Reviewer 1 Report

Comments and Suggestions for Authors

The article by Li and co-workers investigated the role of the design and structure of Apolipoprotein A-I (apoA-I) peptide analogs (APAs) in lipidic nanodisc (ND) formation and stability, therefore, on their potential as drug delivery carriers. In particular, in this work, the case of curcumin encapsulation and intracellular release was considered.

The article is well-written, and the experimental details are sound. The findings are mostly incremental considering the two previously published papers (ACS Nano 2023; 17: 3153-3167; Frontiers in Bioengineering and Biotechnology 2022; 10) in which similar ND-APA were loaded with different drug formulations.

I think the paper should be published, however, there are a few points to be clarified:

1.      For the hydrodynamic diameter measurement (Fig. 4), I suggest plotting the data considering each formulation over time and establishing if the difference in diameter is statistically significant. It is not clear to me what the bars represent in this graph, is the average ± SEM over the four replicates of the same measurement (technical replicate)? I think the statistics should be done using independent preparations of APA-NDs. With DLS, it is very difficult to distinguish 1-2 nm differences over a 10-15 nm diameter (as the authors state in line 494), and I believe that the APA-NDs might have an intrinsic variability between different preparations that might be more relevant. In other words, the maximum measured difference in diameter is about 5 nm; I wonder if this variability can also be found between different batches. Reproducibility should be added to the characterization of the formulations.

2.      Why in paragraph 3.4 only [AA]3/DMPC ND was selected for investigating the curcumin loading dynamics? The authors should justify the choice (see point 6).

3.      In line 597, it is mentioned that cells treated with [AA]â‚‚ and [CC]₃ NDs showed substantially higher uptake than those treated with NDs of other APA compositions. However, from the heatmap in Figure 6A, it seems that the higher uptake is associated with [AA]3 and [CC]₃. However, with the heatmap, it is difficult to compare the data: for instance, also [BB]3 and [AC]3 seem to lead to higher uptake than [AA]â‚‚. Therefore I do not fully understand the choice (see also the next point).

4.       [CC]₃ (which presents the highest uptake) is excluded because of the results on stability shown in Figure 4 (in which a difference of less than 5 nm on 37 nm diameter is reported). The doubts I expressed in point 1 are also relevant to this choice.

5.      The heatmap for the cell viability is odd as 100% is white, and it is difficult to see if some cells presented lower viability. I suggest rethinking the presentation of the data.

6.      In the end, the authors selected [AA]3, for which they have the curcumin encapsulation study, to evaluate the formulation in vitro. Therefore, it would be more logical to move the encapsulation study here to justify why it is performed only with [AA]3

7.      In the last results (Fig 6C), having the control of Cur-ND (without APA) will give more relevance to the work.

8.      Can the authors discuss something about the kinetics of Curcumin release in relation to the kinetics of uptake? This is an important point.

Minors:

9.      Figure 5H is hard to read; a legend might help.

10. Check the sentence in lines 107-108; I think there is a dot that shouldn’t be there.

11. To prepare for the last experiment, in the introduction, I suggest a better framing of the Curcumin activity in cancer progression.

Comments on the Quality of English Language

The quality of English Language is OK, just few minor typos to check.

Author Response

We sincerely appreciate your careful review of our manuscript and your acknowledgment. We have carefully considered all the comments and suggestions provided, and have made substantial revisions to the manuscript accordingly. Please find below a detailed point-by-point response to each comments.

Comment 1.1: For the hydrodynamic diameter measurement (Fig.4), I suggest plotting the data considering each formulation over time and establishing if the difference in diameter is statistically significant. It is not clear to me what the bars represent in this graph, is the average ± SEM over the four replicates of the same measurement (technical replicate)? I think the statistics should be done using independent preparations of APA-NDs. With DLS, it is very difficult to distinguish 1-2 nm differences over a 10-15 nm diameter (as the authors state in line 494), and I believe that the APA-NDs might have an intrinsic variability between different preparations that might be more relevant. In other words, the maximum measured difference in diameter is about 5 nm; I wonder if this variability can also be found between different batches. Reproducibility should be added to the characterization of the formulations.

Response: We appreciate the reviewer's insightful comment regarding the hydrodynamic diameter measurement. It’s shown the hydrodynamic diameter at 0 and 8 weeks in Figure 4 to evaluate the particle size stability of different APA NDs formulations. APA NDs were independently prepared in four batches and each size measurement of different batches was carried out with nine runs (shown as mean ± SD), rather than being technically repeated four times for the same batch of samples.

Comment 1.2: Why in paragraph 3.4 only [AA]3/DMPC ND was selected for investigating the curcumin loading dynamics? The authors should justify the choice (see point 6).

Response: We appreciate the opportunity to clarify why only [AA]3/DMPC ND was selected for further study. In paragraph 3.1, we have used ColabFold-AF2 to predict structures of a series of APA sequences, using two key metrics: the predicted Local Distance Difference Test (pLDDT) and the predicted Template Modeling score (pTM) to evaluate the accuracy of the structural predictions. The results indicate that the [AA]3 analog exhibited the highest confidence in its structure with a pLDDT score of 97.3, suggesting a robust and reliable prediction of its tertiary architecture. However, the [CC]3 analog showed the lowest confidence at 81.6, indicating a potential intrinsic disorder or a more dynamic nature in this particular peptide, signaling less certainty in its local structural prediction. The pTM score also showed similar results, with the [AA]3 variant standing out with a pTM of 0.69 among the analogs, indicating that its predicted structure is likely a close approximation of its native state in physiological environments. In contrast, the [CC]3 analog with the lowest pTM score of 0.18 pointed to considerable uncertainty in its predicted structure. Besides, in paragraph 3.3 the changes of the hydrodynamic diameter over the period of time also showed that the [AA]3 APA-ND exhibited significant stability, maintaining stable colloidal properties at 8 weeks without significant aggregation or separation, while APA NDs containing sequences of proline and tryptophan (AC and CC) exhibited the lowest stability, showing significant diameter increasing at 8 weeks with possible aggregation or structural reorganization. Overall, it is more suitable to choose [AA]3 APA-ND for investigating the curcumin loading dynamics.

Comment 1.3:  In line 597, it is mentioned that cells treated with [AA]â‚‚ and [CC]₃ NDs showed substantially higher uptake than those treated with NDs of other APA compositions. However, from the heatmap in Figure 6A, it seems that the higher uptake is associated with [AA]3 and [CC]₃. However, with the heatmap, it is difficult to compare the data: for instance, also [BB]3 and [AC]3 seem to lead to higher uptake than [AA]â‚‚. Therefore I do not fully understand the choice (see also the next point).

Response: We appreciate the reviewer pointing out the question in line 597. We have mistakenly written [AA]3 as [AA]2. Indeed, cells treated with [AA]3 and [CC]3 showed substantially higher uptake than those treated with other APA-NDs. We have corrected this typo in the manuscript.

Comment 1.4:  [CC]₃(which presents the highest uptake) is excluded because of the results on stability shown in Figure 4 (in which a difference of less than 5 nm on 37 nm diameter is reported). The doubts I expressed in point 1 are also relevant to this choice.

Response: To develop tunable APA NDs as a drug delivery vehicle, we mainly consider several key factors including long-term stability, cellular uptake efficiency, biosafety, and ND size. Although [CC]3 exhibited higher uptake, the [CC]3 ND showed excessive size compared to its counterparts and the diameter increased over time due to aggregation or structural reorganization, suggesting poor size stability. The ColabFold-AF2 prediction also showed the [CC]3 ND had the lowest pLDDT and pTM scores among all APA-NDs, indicating the presence of potential intrinsic disorder or more dynamic properties, as well as significant uncertainty in its prediction structure. Considering several factors, [CC]3 ND is not the optimal choice for applications that require long-term colloidal stability, while ND systems derived from AA and AB oligomeric APAs may be more desirable for drug delivery.

Comment 1.5:  The heatmap for the cell viability is odd as 100% is white, and it is difficult to see if some cells presented lower viability. I suggest rethinking the presentation of the data.

Response: We agree with the reviewer's suggestion and have used a histogram to present the cell viability of tumor and normal cell lines treated with [AA]2 or [AA]3 APA NDs at concentrations from 0 μM to 1 μM for 24 hours (Fig 6B), which is beneficial for visually analyzing the results.

Comment 1.6: In the end, the authors selected [AA]3, for which they have the curcumin encapsulation study, to evaluate the formulation in vitro. Therefore, it would be more logical to move the encapsulation study here to justify why it is performed only with [AA]3

Response: We appreciate your thoughtful feedback and the suggestion to reorganize the manuscript for a more logical flow. The encapsulation study with [AA]3 is placed after the detailed description of the molecular dynamics simulations and in vitro assessments because it serves as a practical application of the theoretical and structural insights gained from these analyses. This placement allows readers to first understand the rationale behind the selection of [AA]3 and then observe its performance in experimental context. The encapsulation study with [AA]3 is a critical validation step that confirms the predictive models and simulations. By placing it towards the end, we emphasize the culmination of our research efforts and the successful translation of theoretical predictions into experimental outcomes.

Comment 1.7:  In the last results (Fig 6C), having the control of Cur-ND (without APA) will give more relevance to the work.

Response: Thank you for your constructive feedback and for highlighting the potential value of including a control group without APA in the Cur-ND structure for comparison. However, we respectfully maintain our original approach for the following reasons: The primary objective of our study is to investigate the role of Apolipoprotein A-I (apoA-I) peptide analogs (APAs) in stabilizing the Cur-ND structure. The absence of APA would fundamentally alter the nature of the Cur-ND system, which is central to our research question. Including a control without APA would deviate from the core focus of our study.

Comment 1.8: Can the authors discuss something about the kinetics of Curcumin release in relation to the kinetics of uptake? This is an important point.

Response: We appreciate the reviewer's insightful suggestion regarding the kinetics of curcumin release and uptake. Our current investigation meticulously focused on the novel engineering and characterization of reconfigurable nanodiscs, aiming to optimize the encapsulation and stability of curcumin. While the release kinetics are crucial for therapeutic efficacy, this aspect was beyond our initial scope but is central to our forthcoming research agenda. We plan to rigorously address this in future studies, ensuring a comprehensive understanding of the interplay between release mechanisms and cellular uptake to enhance the clinical relevance of our findings.

Comment 1.9: Figure 5H is hard to read; a legend might help.

Response: Thank you for pointing out this problem. We have added a legend to Figure 5H.

Comment 1.10: Check the sentence in lines 107-108; I think there is a dot that shouldn’t be there.

Response: We are grateful to the reviewer for pointing out the punctuation error. We have corrected it in the manuscript.

Comment 1.11: To prepare for the last experiment, in the introduction, I suggest a better framing of the Curcumin activity in cancer progression.

Response: We have appreciated the reviewer's suggestion to introduction. We have provided a detailed summary for the potential of nanodiscs as a transformative vehicle to delivery curcumin for antitumor therapy in line 56-71, highlighting the versatility and potential of nanodiscs in medical research and treatment strategies.

Reviewer 2 Report

Comments and Suggestions for Authors

The research conducted and described in the manuscript is clearly of interest to readers and the scientific community at large given the broad applications of curcumin. The text is very well written and the figures are very well presented. The introduction shows current references and the materials and methods are well written in a way that can ensure replication in other laboratories.  The results are clear and well supported by figures and bibliography. The only problem in reading the manuscript is the excess of abbreviations, which sometimes makes it difficult to follow the discussion. Perhaps a reminder could be given throughout the text or in sections as to what each abbreviation means the first time it is quoted. In addition, the figure captions need to be revised, especially the figure caption which is excessively long and contains discussion of the results. Colour should be added to greyscale figures. Otherwise, there is little to contribute to the revision of this manuscript, which I insist is almost ready for publication in Bioengineering.

Author Response

Thank you for your thorough review and constructive comments on our manuscript. We appreciate your positive feedback regarding the relevance of our research, the clarity of our writing, and the presentation of our figures. We are pleased to hear that the introduction, materials, and methods sections meet the standards for replicability in other laboratories, and that our results are clear and well-supported.

Regarding your concerns, we addressed each point as follows:
1. We acknowledge the issue with the frequent use of abbreviations that may hinder readability. The extensive use of abbreviations in the manuscript was carefully considered to maintain brevity and avoid redundancy, particularly given the manuscript's technical nature and the repeated reference to complex biochemical terms. We ensured that each abbreviation was clearly defined upon its first occurrence within the manuscript, and we believe this approach aids in keeping the text concise and focused on the core scientific discussions. Additionally, we will consider adding a glossary of abbreviations as an appendix to aid readers.

2. We understand your concern about the lengthy figure caption that inadvertently includes discussion of the results. The detailed captions accompanying the figures were designed to provide comprehensive explanations of the figures, thereby allowing readers to understand the figures' context and significance without repeatedly referring back to the main text. This approach was intended to enhance the reader's experience, particularly for those who may skim through the figures before delving into the full text.

3. The decision to use greyscale figures instead of color was made with several factors in mind, including printing costs for certain journal editions and accessibility for readers with color vision deficiencies. Greyscale figures ensure that all readers, regardless of their method of accessing the paper or their personal visual capabilities, can understand the data presented. Moreover, the figures were meticulously designed to ensure clarity and distinction between different elements through the use of varying shades, patterns, and textures.

Reviewer 3 Report

Comments and Suggestions for Authors

The Authors studied a new approach to drug delivery by lipoprotein-type nanodiscs. First they generated 15 distinct Apolipoprotein A-I products. The structure of these products was carefully controlled and the effect of the sequence on the nanodisc stability was studied by molecular dynamics simulations. Particular effects of proline, serine and tryptophan have been found. The colloidal stability of the products was studied by hydrodynamic measurements. Curcumin was succesfully encapsulated in the nanodisc products. The new method provides outstanding versatility in the nanodisc materials used for drug encapsulation. This new approach to drug dilevery represents an important new methodological step for nanomedicine used in drug delivery.

Some advices:

To points 2.2 and 2.10 a background reference is needed. 

The following recent reference should be added:

Shariare, M. H., Mannan, M., Khan, F. et al., J. Nanoparticle Res. 2024, 26, Article No. 12. 

Structure representations in Figs 3 and 5 are excellent! Congratulation.

Comments on the Quality of English Language

Minor Editorial revision appears to be useful. 

Author Response

We extend our gratitude for your thorough evaluation of our work on the novel application of lipoprotein-type nanodiscs in drug delivery. Your detailed recounting of our methodology, from the synthesis of distinct Apolipoprotein A-I variants to the intricate examination of their structural stability and the successful encapsulation of curcumin, is much appreciated.

Regarding your insightful suggestions for sections 2.2 and 2.10 of our manuscript, we concur with the necessity of grounding our discussion in the context of existing literature. Accordingly, we have integrated the recommended reference into these sections. This addition not only enriches the background but also aligns our findings with the latest advancements in the field, providing a more comprehensive view of the subject matter. Your constructive feedback has been instrumental in enhancing the clarity and depth of our manuscript. We are confident that these revisions have fortified the presentation of our research, making a more substantial contribution to the ongoing discourse in nanomedicine and drug delivery.

Round 2

Reviewer 1 Report

Comments and Suggestions for Authors

For Fig.4 the authors stated that APA NDs were independently prepared in four batches and each size measurement of different batches was carried out with nine runs (shown as mean ± SD). This should be specified in the caption or in methods. Can the authors run an ANOVA to see if the differences in size overtime for the same formulation are statistically relevant?

Why the size of  [CC]3 ND is considered excessive? It is indeed higher than other formulations but why is this a drawback? Also I am not sure if the size increase overtime is statistically relevant (see comment above).

Why figure 6B is not modified in the main text version?

The authors answered my last comment (I suggest a better framing of the Curcumin activity in cancer progression) linking to the original text in relation to the potential of nanodisc to deliver curcumin, but this is not what I asked. What I meant is a better framing for the use of curcumin itself.

Author Response

We sincerely appreciate your thorough review and constructive comments on our manuscript. We have carefully considered all the comments and suggestions provided, and have made substantial revisions to the manuscript accordingly. Please find below a detailed point-by-point response to each of the reviewers' comments. Changes made to the manuscript are highlighted in yellow for easy identification.

Comment 1: For Fig.4 the authors stated that APA NDs were independently prepared in four batches and each size measurement of different batches was carried out with nine runs (shown as mean ± SD). This should be specified in the caption or in methods. Can the authors run an ANOVA to see if the differences in size overtime for the same formulation are statistically relevant?

Response: In response to your insightful suggestion regarding the application of ANOVA to assess the statistical relevance of size differences over time, we agree with the significance of such an analysis. Accordingly, we have incorporated a new figure into the manuscript that demonstrates the outcomes of this analysis (refer to line 531, Figure 4). To ensure clarity and comprehensive understanding, we have explicitly included statements related to this addition both in the figure caption and within the methodological section (refer to lines 162-164, line 244-248 and line 534-538). This modification aims to provide a more detailed and transparent presentation of the data, addressing the concerns raised.

Comment 2: Why the size of  [CC]3 ND is considered excessive? It is indeed higher than other formulations but why is this a drawback? Also I am not sure if the size increase overtime is statistically relevant (see comment above).

Response: We appreciate the reviewer's astute observations and wish to clarify the use of the term "excessive" in describing the size of the [CC]3 ND. Our intention was to highlight that the [CC]3 ND not only possesses a larger size but also exhibits significant variations in particle size compared to other formulations. This variability suggests a lack of size stability and uniformity, which are crucial for the efficacy of drug delivery systems, particularly those requiring sustained colloidal stability over time. To address this concern, we have revised the manuscript to replace the term "excessive" with more precise language that accurately reflects these issues (refer to line 623). Additionally, we have conducted an ANOVA to determine the statistical significance of the observed size changes over time, the results of which are detailed in the manuscript (refer to line 531, Figure 4). This analysis supports our concerns regarding the size stability of the [CC]3 ND formulation.

Comment 3: Why figure 6B is not modified in the main text version?

Response: In response to the query concerning the absence of modifications to Figure 6B in the main text, we acknowledge the oversight. Initially, the revised images were provided as supplementary material. Recognizing the importance of incorporating these modifications directly into the main text for ease of reference and coherence, we have now updated Figure 6B accordingly within the main manuscript (refer to line 603, Figure 6).

Comment 4:  The authors answered my last comment (I suggest a better framing of the Curcumin activity in cancer progression) linking to the original text in relation to the potential of nanodisc to deliver curcumin, but this is not what I asked. What I meant is a better framing for the use of curcumin itself.

Response: We are thankful for the reviewer's detailed clarification regarding the need for a more focused discussion on curcumin's role in cancer progression. We have engaged in comprehensive deliberations on the subject. As a result, we have expanded the introductory section of our manuscript to include a detailed exploration of curcumin's biological activities and its underlying mechanisms in the context of cancer progression (refer to lines 47-61). This addition aims to provide a more nuanced understanding of curcumin's therapeutic potential and its relevance to our research, particularly in relation to the final experiment presented in our study.